Role of glycoproteins in hepatic lipid metabolism: emerging diagnostic markers and therapeutic targets

Lan Bo 1
Huang Sixing 1
Luo Jinping 1
Shi Ping 2 shiping_2025@163.com
Qiang Zhe 1 3 qiangzhe@cqctcm.edu.cn
1 Sichuan-Chongqing Joint Key Laboratory of New Chinese Medicine Creation Laboratory, Chongqing Academy of Chinese Materia Medica, Chongqing College of Traditional Chinese Medicine , Chongqing , China
2 School of Chinese Medicine, Chongqing College of Traditional Chinese Medicine , Chongqing , China
3 College of Pharmacy, Chongqing Medical University , Chongqing , China
Uversky Vladimir
Electronic publication date: 2025 Jul 4
Publication date: 2025
Volume: 13
Electronic Location ID: e19627
Received 2024 Oct 4; Accepted 2025 May 30
Copyright: © 2025 Lan et al.
Copyright year: 2025
Copyright holder: Lan et al.
License: This is an open access article distributed under the terms of the Creative Commons Attribution License, which permits unrestricted use, distribution, reproduction and adaptation in any medium and for any purpose provided that it is properly attributed. For attribution, the original author(s), title, publication source (PeerJ) and either DOI or URL of the article must be cited.
License URL: https://creativecommons.org/licenses/by/4.0/

Keywords: Glycoprotein, Hepatic lipid metabolism, Diagnostic marker, Cancer biomarker

Funding: Chongqing Municipal Performance Incentive Guidance Special cstc2023jxjl-jbky130011 Key Programs of Technological Innovation and Application Development of Chongqing, China cstc2021jscx-dxwtBX0016, CSTB2023TIAD-KPX0101 Natural Science Foundation of Chongqing, China cstc2021jcyj-msxmX0793 Chongqing Municipal Education Commission KJQN202215126 This work was funded by Chongqing Municipal Performance Incentive Guidance Special (grant number, cstc2023jxjl-jbky130011), Key Programs of Technological Innovation and Application Development of Chongqing, China (grant number, cstc2021jscx-dxwtBX0016; CSTB2023TIAD-KPX0101), Natural Science Foundation of Chongqing, China (grant number, cstc2021jcyj-msxmX0793) and Science and Technology Research Program of Chongqing Municipal Education Commission (grant number, KJQN202215126). The funders had no role in study design, data collection and analysis, decision to publish, or preparation of the manuscript.

==============================
The liver is an important metabolic organ in the human body, and abnormal hepatic lipid metabolism may lead to a variety of diseases such as metabolic dysfunction-associated fatty liver disease, cirrhosis, and hepatocellular carcinoma. These diseases are often closely associated with chronic diseases, including obesity, hypertension, high cholesterol and diabetes, and have become major factors increasing the global disease burden. Currently, the diagnosis of diseases related to hepatic lipid metabolism relies mainly on imaging findings and serologic markers. In terms of treatment, lifestyle interventions such as dietary changes and increased exercise remain the mainstay of the fight against the diseases. However, effective therapies to control the progression of these diseases are still lacking. Glycoproteins are complex biomolecules consisting of sugar chains and proteins bound covalently, which are widely found in cell membranes and secretory fluids. In recent years, researchers have found that glycoproteins have potential application value in the diagnosis and treatment of diseases related to hepatic lipid metabolism. But there are currently no articles summarizing and analyzing the pathways and mechanisms involved. This review provides an overview of the progress of research on glycoproteins involved in diseases related to hepatic lipid metabolism as well as new insights into glycoproteins as potential diagnostic markers and therapeutic targets for lipid metabolism diseases.

Introduction

With the increase of obesity, diabetes, and metabolic syndrome globally, liver diseases caused by lipid metabolism disorders have become a public health concern worldwide (Stefan, Häring & Cusi, 2019). Abnormal lipid metabolism induces fat deposition in the liver, which can trigger conditions such as non-alcoholic fatty liver, non-alcoholic steatohepatitis, liver fibrosis, and cirrhosis. Hepatic lipid metabolism disorder is closely related to factors such as obesity, excessive fat intake, lack of exercise, genetic factors, and insulin resistance, which result in excessive hepatic fatty acid synthesis and interference with uptake and oxidation, leading to fat accumulation in the liver (Ferguson & Finck, 2021). Identifying effective diagnostic biomarkers and therapeutic targets to improve early diagnosis and treatment outcomes will be the focus of future research.

Glycoproteins are biopolymers composed of proteins and sugar chains, possessing various important biological functions and playing a key role in disease diagnosis and treatment. The expression patterns of glycoproteins may serve as a tumor marker, assisting physicians in diagnosing malignant tumors. By detecting abnormal expression of glycoproteins in the body, the presence of tumors can be determined at an early stage (Llop et al., 2018). Additionally, glycoproteins can also act as diagnostic markers of inflammatory diseases, heart diseases, and autoimmune diseases, guiding physicians in intervention and treatment in a timely manner (Al-Ani, Chen & Costello, 2023; Núñez et al., 2021; Hu et al., 2019; Camilli et al., 2022).

Glycoproteins are also important in the modulation of liver lipid metabolism. By interacting with lipid-synthesizing enzymes and membrane proteins, glycoproteins are involved in the synthesis, metabolism, and transport of lipids in the liver. Furthermore, glycoproteins affect the degradation rate of lipids, thereby maintaining the balance of lipid metabolism and playing a critical regulatory role in hepatic lipid metabolism (Min-DeBartolo et al., 2019; Zhang et al., 2017; Islam et al., 2022). Glycoproteins play an important role in the diagnosis and treatment of diseases related to hepatic lipid metabolism. By studying the expression characteristics and functions of glycoproteins, we can provide more accurate and effective means for early diagnosis and treatment of diseases. This review focuses on the study of the role and regulatory mechanisms of glycoproteins in the development of liver lipid metabolism-related disorders (Fig. 1), previewing that glycoproteins can provide more accurate and effective means for early diagnosis and treatment of hepatic diseases in the future.

Figure 1 Diagram for the main content organization of this review.

PEDF, pigment epithelium-derived factor; ZAG, Zinc-α2-glycoprotein; SR-BI, scavenger receptor class B type I; L-PGDS, Lipocalin-type prostaglandin D2 synthase.

Survey methodology

The literature search was conducted in PubMed, Google Scholar and Web of Science resources. Research and review articles published since 2012 (prior to this, studies on the mechanisms involved in the regulation of hepatic lipid metabolism by glycoproteins had not emerged) in the English language were only included in this review. The search strategy included the following search terms and Boolean operators using the term “glycoprotein” AND “hepatic lipid metabolism”, “glycoprotein” AND “lipid metabolism related diseases”, OR “regulation of lipid metabolism”, OR “biomarker”, OR “therapeutic strategy”, OR “therapeutic target”, OR “lipotoxicity”, OR “hepatocyte steatosis”, OR “inflammation”. After removing duplicate articles and the articles with little relevance, 94 articles were selected for inclusion in this review. Doctors and researchers specializing in hepatobiliary diseases, endocrine medicine and clinical laboratory medicine will find the review of current scientific literature of particular interest. In addition, this review is also suitable for the journal’s readership engaged in basic medical research.

Glycoproteins modulate lipid metabolism

The balance of lipid metabolism is the basis for maintaining the body’s energy metabolism and cellular homeostasis. Disruption of this process leads to the occurrence of various diseases, including atherosclerosis, obesity, diabetes, and cancer (Yoon et al., 2021). Glycoproteins, constituting a major class of proteins, are important regulators of the complex lipid metabolism balance in the body. Cluster of Differentiation 36 (CD36) is a transmembrane glycoprotein that, as part of the subcellular vesicle cycle, promotes the uptake of long-chain fatty acids by cells, thereby regulating its presence on the cell surface and playing essential roles in the pathogenetic mechanisms of insulin resistance and diabetic cardiomyopathy (Glatz & Luiken, 2018). Pigment epithelium-derived factor (PEDF) is another multifunctional secretory glycoprotein that exerts a protective effect on hypoxia-induced cell death and controls lipid metabolism by inducing autophagy in hypoxic H9c2 cells (Zhang et al., 2018). The membrane glycoprotein SR-B1, which is highly homologous to CD36, was firstly detected on the membrane of hepatocytes, mediating the recognition and absorption of high-density lipoprotein (HDL) in the liver. However, when expressed by vascular endothelial cells, SR-B1 actively absorbs low-density lipoprotein (LDL) in the blood, playing a pathogenic role in atherosclerosis (Wang et al., 2018; Huang et al., 2019). Zinc-α2-glycoprotein (ZAG), stimulating lipid mobilization, was shown to alleviate the reduction of blood lipid levels induced by dexamethasone, suggesting its potential beneficial role in metabolic stress (Qiao et al., 2019). B7-H3 overexpression in lung cancer leads to abnormal lipid metabolism through the SREBP-1/FASN axis, resulting in poor prognosis of lung cancer (Luo et al., 2017). Lipocalin-type prostaglandin D2 synthase (L-PGDS), a glycoprotein expressed in various tissues, is related to multiple metabolic disorders, including diabetes and fatty liver, suggesting an important role for this glycoprotein in cancer-related metabolism (Islam et al., 2022). Secretory glycoprotein angiopoietin-like 4 (ANGPTL4), a regulator of lipid metabolism, has also been found to participate in the growth and metastasis of osteosarcoma (OS) through activation of the mTOR signaling pathway (Lin et al., 2022).

These studies highlight the different roles played by specific glycoproteins in lipid metabolism and the molecular mechanisms by which glycoproteins regulate lipid metabolism and thus affect the development of a variety of diseases. The new insights and mechanisms may help reveal the potential links between glycoproteins and metabolic diseases and provide effective drug targets and therapeutic or prevention strategies for related disorders.

Involvement of glycoproteins in diseases related to hepatic lipid metabolism

Glycoproteins are widely implied in body lipid metabolism, and the liver, as the most important metabolic organ in the body, is closely related to lipid metabolism. Therefore, a variety of glycoproteins contribute to different processes, e.g., hepatic lipid synthesis, catabolism, and transport, which have important impacts on numerous diseases related to hepatic lipid metabolism (Fig. 2).

Figure 2 The pathways and mechanisms about glycoproteins in liver metabolic disturbance.

SHBG, sex hormone-binding globulin; ZAG, Zinc-α2-glycoprotein; SR-BI, scavenger receptor class B type I. (Created with MedPeer).

Zinc alpha2 glycoprotein

Zinc α2-glycoprotein (ZAG) modulates lipolysis and substantially affects fat metabolism. However, how ZAG affects liver lipid metabolism remains largely undefined. Studies on animal models have shown that ZAG significantly reduces the body weight and liver triglyceride (TG) content in mice with high-fat diet-induced obesity. This was accompanied by a significant downregulation of proteins controlling lipid synthesis, e.g., stearoyl-CoA desaturase 1 (SCD1) and acyl-CoA synthetase-1 (ACSS1) (Fan et al., 2017). Further studies have found that ZAG overexpression can alleviate palmitate-induced intrahepatic lipid accumulation by inhibiting fat synthesis and promoting lipolysis and fatty acid β-oxidation. These findings suggest that ZAG has an important regulatory function in liver lipid accumulation (Xiao et al., 2017).

Recently, ZAG overexpression was shown to upregulate adiponectin and lipolysis genes, while downregulating fat synthesis genes, thus preventing NAFLD by alleviating hepatic steatosis, insulin resistance, and inflammation (Xiao et al., 2018). ZAG also reduces NAFLD by negatively regulating tumor necrosis factor-α (TNF-α), indicating that ZAG can exert a protective effect on non-alcoholic fatty liver disease (NAFLD) by modulating various metabolic and inflammatory pathways, demonstrating ZAG’s multifaceted role in lipid metabolism and inflammation (Namkhah et al., 2021; Liu et al., 2019). Despite increasing evidence highlighting ZAG’s role in liver lipid metabolism, its exact molecular mechanisms and therapeutic potential remain unclear. Further research is warranted to provide a deeper understanding of ZAG’s impact on liver lipid metabolism and its potential as a therapeutic target.

CD36

CD36, a lipid transport protein, primarily regulates the body’s lipid metabolism by mediating lipid uptake and transport. The CD36-PPAR pathway is a recognized regulator of glucose and energy metabolism and is associated with various metabolic disorders by promoting long-chain fatty acid uptake (Glatz & Luiken, 2018; Yu et al., 2021). Tenofovir disoproxil fumarate (TDF), a drug used in HBV infection, was shown to regulate lipid metabolism through activation of hepatic CD36-PPARα pathway and is linked to atherosclerosis and liver cancer development (Suzuki et al., 2021). The PPARγ-CD36 pathway is involved in NAFLD development. CD36 is considered an important factor in NAFLD development. Given that CD36’s palmitoylation status is associated with disrupted fatty acid metabolism and tissue inflammation, inhibiting CD36 palmitoylation could represent a potential treatment strategy for NAFLD (Maréchal et al., 2018; Ding et al., 2019; Zhao et al., 2022). CD36 and growth hormone-releasing peptides, such as hexarelin, activate PPAR-γ and may serve as new therapeutic targets for regulating lipid and energy metabolism (Hao et al., 2018). Additionally, CD36-mediated long-chain fatty acid uptake activates hepatocytes and promotes oxidative stress, thus contributing to liver fibrosis development (Shirpoor et al., 2018; Xu et al., 2018). The above data emphasize the significance of understanding the mechanisms by which CD36 and PPARs regulate hepatic lipid metabolism and provide valuable insights into the management of liver lipid metabolic disorders.

Sex hormone-binding globulin

Sex hormone-binding globulin (SHBG), an important glycoprotein produced and secreted by the liver, specifically binds and transports sex hormones. It can bind to testosterone, estradiol, insulin, thyroid hormones (T3, T4), and other sex hormones, thus controlling the availability of these hormones to body cells (Kornicka-Garbowska et al., 2021). Recently, reports have increasingly explored the complex regulatory mechanisms of SHBG in endocrine functions and its role in metabolic syndromes, e.g., obesity, diabetes mellitus, and NAFLD, highlighting its potential as a diagnostic marker and therapeutic target (Simons et al., 2021). Low serum SHBG levels are often detected in patients with metabolic disorders, indicating close associations of SHBG with insulin resistance and lipid dysregulation. Other studies have suggested an intrinsic mechanistic link between SHBG and metabolic syndrome. For example, SHBG levels are negatively correlated with markers of hepatic de novo lipogenesis (DNL), indicating that enhanced lipogenesis contributes to reduced SHBG synthesis and release, especially in individuals with obesity and hepatic steatosis. This inverse relationship is particularly pronounced in women, suggesting sex-specific regulatory mechanisms (Kornicka-Garbowska et al., 2021; Simons et al., 2021; Zhang et al., 2022). Furthermore, the association between low SHBG levels and NAFLD has been confirmed by numerous studies (Zhang et al., 2022; Lee et al., 2019; Saez-Lopez et al., 2017; Luo et al., 2018). Notably, these studies emphasize that low SHBG expression is a common characteristic of NAFLD patients (Zhang et al., 2022; Lee et al., 2019; Luo et al., 2018). SHBG is involved in lipogenesis regulation, particularly through enzymes such as acetyl-CoA carboxylase (ACC), highlighting its functional significance in liver lipid metabolism. This underscores the potential use of SHBG as a molecular marker for early identification of individuals with high odds of NAFLD (Saez-Lopez et al., 2017; Sáez-López et al., 2019; Sarkar et al., 2019).

SHBG and hepatocellular carcinoma: The protective role of SHBG also has potential implications for liver cancer development. In postmenopausal women, SHBG appears to inhibit NAFLD-induced hepatocellular carcinoma (HCC) through regulation of estrogen activity and inhibition of lipogenesis (Lee et al., 2019). Transgenic mouse models were used to provide insights into SHBG’s ability to mitigate liver tumor progression under high-fat diet conditions, further supporting its therapeutic prospects.

SHBG in special populations: The impact of altered SHBG levels is particularly evident in specific populations, such as adolescents with polycystic ovary syndrome (PCOS) and individuals with glycogen storage disease type 1a (GSD1a). In adolescents with PCOS, decreased SHBG levels are a concerning indicator of high NAFLD risk, exacerbated by obesity and metabolic disorders (Urbano et al., 2022). Similarly, in GSD1a patients, the genetically driven increase of DNL leads to significantly reduced SHBG levels, confirming the inverse relationship between lipogenesis and SHBG expression (Simons et al., 2022).

Despite increasing research evidence, the precise molecular mechanisms linking SHBG downregulation to liver lipid metabolism-related diseases remain unclear. The inverse relationship between SHBG and hepatic lipogenesis and its strong association with NAFLD emphasize its diagnostic and therapeutic potential. With further under-standing of SHBG’s mechanisms, targeted regulation of its expression may provide new approaches for managing metabolic liver diseases.

Lactoferrin and Sidt2

Lactoferrin, a critical iron-binding glycoprotein, is found in human milk and various secretions, serving as a core immune function protein in breast milk. It has gained attention in recent years for its potential therapeutic role in lipid metabolism and liver health. Lactoferrin regulates cholesterol metabolism in the liver and effectively reduces hepatic cholesterol levels by modulating synthetic and transport path-ways, thereby maintaining liver health (Abd El-Hack et al., 2023). It also regulates lipid metabolism via suppression of the farnesoid X receptor (FXR)-mediated gut-liver axis, reducing hepatic cholesterol deposition and enhancing bile acid metabolism (Ling et al., 2019). Additionally, lactoferrin alleviates inflammation and oxidative stress, improves diet-induced hepatic steatosis and inhibits liver lipogenesis (Xiong et al., 2018).

Metformin is known to activate AMP-activated protein kinase (AMPK), thereby sustaining cellular energy homeostasis and inhibiting hepatic lipogenesis (Viollet et al., 2012). Animal studies have demonstrated that metformin alone significantly reduces lipid accumulation and improves hepatic steatosis (Wang et al., 2021). Recent studies on combined treatment with lactoferrin and metformin have further revealed potential synergistic effects for these compounds. Indeed, combined administration of metformin and lactoferrin (Met+Lf) significantly reduced body weight, visceral fat, and serum triglycerides in mice fed a high-fat diet; furthermore, combined treatment was shown to enhance AMPK activation, inhibit lipogenic enzyme expression, and upregulate lipolytic enzymes, suggesting potential comprehensive mechanisms involving improved lipid metabolism and energy expenditure (Li et al., 2022).

Although these findings are promising, there are still gaps in understanding the regulation of liver lipid homeostasis, autophagy, and inflammatory pathways. Additionally, while the lysosomal membrane protein Sidt2 is associated with lipid metabolism and NAFLD, how Sidt2 affects hepatocyte lipid metabolism, particularly through autophagy, remains unclear (Chen, Gu & Zhang, 2018). Sidt2 knockout models exhibit significant lipid metabolic disorders and spontaneous NAFLD development, accompanied by transmissible endoplasmic reticulum stress (TERS), suggesting potential interactions with lactoferrin’s lipid-regulating pathways (Gu et al., 2021; Qian et al., 2023; Moon et al., 2018). Understanding the interactions of lactoferrin, lipid metabolism, and key regulatory proteins such as Sidt2 may help further enhance the therapeutic potential of lactoferrin and improve its clinical applicability.

Other glycoproteins associated with hepatic lipid metabolism

The mechanism of hepatic steatosis is impacted by multiple factors and different physiological processes involving various glycoproteins and signaling pathways. As research progresses, glycoproteins are continually found to be involved in the development of hepatic lipid metabolism-related diseases. Traditional models have focused on intrinsic cellular pathways within individual organs. However, current under-standing has evolved to recognize the importance of inter-organ communication facilitated by a variety of secreted factors known as organoids.

Liver-adipose tissue axis: Glycoprotein non-metastatic melanoma protein B (GPNMB), secreted by the liver, has been identified as a key facilitator of adipogenesis in white adipose tissue (WAT), exacerbating obesity and insulin resistance. Inhibiting GPNMB may improve these conditions and promote WAT formation (Gong et al., 2019).

SR-BI and lipid metabolism: SR-BI regulates the plasma levels of apolipoprotein E (Apo E) and dietary lipid deposition in the liver, affecting de novo fatty acid synthesis and lipid uptake. Animal studies showed that SR-BI-deficient mice exhibit significant resistance to diet-induced obesity and liver lipid deposition, indicating that class B type I scavenger receptor (SR-BI) might be crucial for liver lipid regulation (Karavia et al., 2015).

Role of melatonin: Melatonin, a circadian rhythm regulator, also has an essential function in glucose and lipid metabolism. It improves insulin resistance and hepatic steatosis by attenuating α-2-HS-glycoprotein (AHSG), suggesting its involvement in the pathogenetic mechanisms of diabetes and NAFLD (Heo et al., 2018).

Gp78: Gp78 is associated with lipid accumulation in hepatocytes; its overexpression induces steatosis, characterized by increased lipid accumulation in the liver. Conversely, Gp78 downregulation mitigates these effects, highlighting its regulatory role in hepatic lipid metabolism (Li et al., 2017).

Skeletal muscle secreted protein (SCL): Elevated levels of circulating SCL, a novel myokine, are linked to insulin resistance in skeletal muscle and liver. SCL up-regulates endoplasmic reticulum (ER) stress through mTOR-mediated autophagy inhibition, impairing insulin signaling and promoting hepatic steatosis (Oh et al., 2022).

N-glycosylation of SCAP: Aberrant N-glycosylation of SREBP cleavage-activating protein (SCAP) exacerbates lipid accumulation and liver inflammation by enhancing histone H3 K27 acetylation mediated by acetyl-CoA synthetase 2 (ACSS2), leading to non-alcoholic steatohepatitis (NASH) (Li et al., 2024).

Extracellular matrix (ECM) and various secreted glycoproteins have also been shown to be key regulators in metabolic liver diseases:

Thrombospondin-1 (TSP-1): TSP-1, an ECM glycoprotein, highly regulates hepatic stellate cell activation and fibrosis in NAFLD/NASH; meanwhile, TSP-1 deficiency prevents various NASH phenotypes (Min-DeBartolo et al., 2019).

Olfactomedin 2 (OLFM2): This adipose tissue-derived glycoprotein modulates hepatic lipid metabolism and mediates significant interactions between subcutaneous adipose tissue (SAT) and the liver, emphasizing its importance in NAFLD pathogenesis (Barrientos-Riosalido et al., 2023).

β-Cystine (cysteine-rich acidic secretory protein): β-Cystine affects adipogenesis, ECM regulation, and metabolic pathways associated with obesity and NAFLD. Evidence suggests that β-cystine can be used as a molecular marker and therapeutic target in metabolic diseases (Atorrasagasti, Onorato & Mazzolini, 2023).

These studies elucidated inter-organ communication mechanisms and the specific roles of various glycoproteins in diseases related to hepatic lipid metabolism, particularly secreted glycoproteins and their regulatory roles in the progression of liver lipid metabolic diseases. Understanding these complexities may help develop new therapeutic pathways targeting inter-organ communication, potentially altering treatment outlook for obesity, T2D, and NAFLD.

Glycoproteins and diagnostic markers

Glycoproteins play key roles in cell signaling, immune response, and cell recognition, attracting increasing attention from researchers as diagnostic markers in recent years. Related studies across multiple fields have been performed, examining neuro-degenerative, cardiovascular and malignant diseases.

Neurodegenerative diseases

Alzheimer’s disease (AD) features a gradual loss of neurons and cognitive decline. Despite progress in understanding the pathophysiology of AD, the precise molecular mechanisms remain elusive. Current evidence indicates that activated microglia play a crucial role in the onset and progression of AD. Glycoprotein non-metastatic melanoma B (GPNMB) was identified as a new marker of activated microglia in AD, with significantly increased expression in transgenic AD models and AD patients. GPNMB co-localizes with IBA1-positive microglia near amyloid plaques, and its expression is directly affected by soluble amyloid β-protein (Aβ) in vitro. This finding positions GPNMB as a potential biomarker of AD and suggests its involvement in dis-ease-associated neuroinflammatory responses (Hüttenrauch et al., 2018).

Cardiovascular and inflammatory diseases

Cardiovascular diseases accompanied by complications such as end-stage renal disease (ESRD) and antiphospholipid syndrome (APS) are serious challenges in clinical management. Studies have demonstrated that the β2-glycoprotein I/IgA immune complex (B2A-CIC) predicts thrombus formation in APS patients post-kidney transplantation. Patients positive for IgA aB2GP1 and B2A-CIC have markedly elevated incidence of systemic and graft thrombosis compared with those negative for these markers. This finding suggests that B2A-CIC may serve as a critical predictor of thrombotic events (Serrano et al., 2017). In ESRD patients, zinc-α2-glycoprotein (ZAG) has emerged as a potential cardiovascular risk biomarker. An observational study reported the prognostic significance of ZAG for cardiovascular events and mortality in dialysis patients (Schmitt, 2018). Similarly, carbohydrate antigen 125 (CA125), or mucin 16, is recognized for its role in heart failure (HF). CA125 levels are closely associated with congestion and inflammation in acute decompensated heart failure syndrome, indicating its reliability as a marker for monitoring and guiding heart failure treatment (Núñez et al., 2021).

Cancer diagnostic markers

Glycoproteins play crucial roles in cancer biology. In cancer, GPNMB is associated with induced epithelial-mesenchymal transition (EMT) and the acquisition of cancer stem cell (CSC) characteristics. Studies using three-dimensional spheroid cultures have shown that GPNMB is expressed on the surface of dormant cancer cells, enhancing their stem cell-like properties. Surface GPNMB may be a marker of and therapeutic target for cancer stem cells (Chen et al., 2018). In hepatocellular carcinoma (HCC), CD133 is another transmembrane glycoprotein whose expression is controlled by transforming growth factor-beta (TGF-β). Specifically, TGF-β upregulates CD133 by inhibiting DNA methyltransferase, thereby enhancing the tumorigenic potential of CD133+ cells. This epigenetic regulation underscores the potential of targeting the CD133 pathway in cancer therapy (You, Ding & Rountree, 2010).

Metabolic and pediatric impacts

Childhood obesity features chronic low-grade inflammation and oxidative stress, associated with endothelial dysfunction. A study investigating urinary biomarkers showed that obese children exhibit elevated levels of inflammatory (CRP, AGP, IL-6) and oxidative stress (8-isoprostaglandin, 8-OHdG) markers, which are correlated with endothelial dysfunction markers. These findings suggest a critical role for non-invasive urinary biomarkers in the early detection and management of cardiovascular risk in obese children (Selvaraju et al., 2019).

These studies highlight the multifaceted roles of glycoproteins in various medical conditions. From biomarkers of neurodegenerative diseases, cardiovascular risk, and cancer stem cell characteristics to their potential as therapeutic targets, glycoproteins provide a promising avenue for future research, which may significantly advance diagnostic and treatment strategies across medical fields.

Glycoprotein markers associated with metabolic liver disease

The progression and complications of diseases associated with hepatic lipid metabolism, including NAFLD, liver fibrosis, cirrhosis, portal vein thrombosis (PVT), and acute liver failure (ALF), often require stringent monitoring. Due to their ease of detection in body fluids, glycoproteins have emerged as promising tools for diagnosis, prognosis, and therapeutic evaluation in these diseases.

Portal vein thrombosis and biliary atresia

Despite lower platelet counts in cirrhotic patients, portal vein thrombosis (PVT) frequently occurs post-hepatectomy and splenectomy. The increased incidence of PVT in cirrhotic patients is related to alterations in platelet activation, with soluble glycoprotein VI (sGPVI) considered a significant marker (Matsui et al., 2018). In biliary atresia, a devastating pediatric liver disease, cartilage oligomeric matrix protein (COMP) was identified as a progressive marker of liver fibrosis. COMP levels correlate with disease severity markers, including jaundice and hepatic dysfunction, and may predict patient survival. The ability of COMP to serve as a non-invasive biomarker could improve the diagnosis and monitoring of disease progression (Udomsinprasert et al., 2021).

Chronic liver disease and liver injury

ZAG expression levels reflect liver and kidney functions in clinical chronic liver disease (CLD) and predict patient survival. High serum ZAG levels are associated with liver function protection and reduced mortality, demonstrating the usefulness of ZAG as a prognostic marker (Shigefuku et al., 2024). GPNMB upregulation in macrophages during liver injury recovery is associated with the severity and prognosis of ALI and ALF. Higher GPNMB levels correlate with poorer outcomes, indicating its potential as a prognostic marker of acute liver disease (Kumagai et al., 2023).

Liver fibrosis and cirrhosis

Procollagen C-proteinase enhancer protein 1 (PCPE-1) was identified as a plasma marker of liver tissue fibrosis. Elevated levels of PCPE-1 parallel fibrosis progression, suggesting its clinical potential in early and mid-stage fibrosis (Hassoun et al., 2016). Reelin is known for its role in brain development. Recent studies have demonstrated its abnormal expression in hepatic stellate cells/myofibroblasts, which is positively correlated with fibrosis stage in HCV-related chronic hepatitis. This indicates that Reelin is involved in the occurrence of liver fibrosis and is expected to be included into the diagnosis and monitoring of liver fibrosis (Carotti et al., 2017). Orosomucoid (ORM) is an acute-phase glycoprotein that plays multiple roles in liver diseases. The different glycosylation forms of ORM may help diagnose acute liver failure (ALF) related to HBV and monitor fibrosis in chronic hepatitis (Elpek, 2021). Lipocalin-2 (LCN-2) has demonstrated superior diagnostic performance for hepatocellular carcinoma (HCC), surpassing traditional biomarkers such as alpha-fetoprotein (AFP). A study involving 300 patients confirmed elevated serum LCN-2 levels in HCC, highlighting its effectiveness in distinguishing cirrhotic disease with and without HCC (Barsoum, Elgohary & Bassiony, 2020).

Hepatitis and liver tumors

In chronic HBV infection, lysosome-associated membrane glycoprotein 3 (LAMP 3) is involved in T cell activation and adaptive immune response. Elevated LAMP 3 expression in the liver is associated with increased T cell infiltration and liver dysfunction (Wang et al., 2023), suggesting LAMP 3 may be a valuable marker of immune dysregulation in HBV infection and disease progression. In patients with liver cirrhosis and cases progressing to HCC, human Mac-2 binding protein glycosylation isomer (M2BPGi) levels are significantly elevated. More importantly, M2BPGi outperforms alpha-fetoprotein (AFP) in predicting the progression to HCC in HBV patients, highlighting M2BPGi as a promising biomarker of liver fibrosis and HCC, especially in patients with chronic HBV and HCV infections (Jun et al., 2019).

As a secretory glycoprotein, cholesterol has shown potential as a diagnostic and prognostic marker of HCC. A study indicated significantly elevated serum levels of clusterin in HCC patients, correlating with disease progression and treatment response (Rasmy et al., 2022), making it a valuable tool for monitoring HCC, especially post-local therapy. Other prognostic markers, e.g., CD147, have also been assessed for their roles in HCC prognosis. Elevated circulating levels of CD147 are associated with poor survival rates in advanced HCC patients, indicating its potential as an independent prognostic marker (Lee et al., 2016). Additionally, clusterin expression in non-tumorous liver tissue correlates with worse post-surgical outcomes, highlighting its potential as a prognostic marker in HCC patients undergoing resection (Kuo et al., 2019).

Moreover, several novel glycoprotein markers have been identified. For instance, α1-acid glycoprotein (AGP) and Gp73 have been examined as potential markers of HCC. Specifically, S2-bound AGP is considered a possible marker for early HCC detection in chronic HCV patients, although its predictive advantages remain to be confirmed (Oltmanns et al., 2024). Meanwhile, Gp73 is associated with the progression of liver fibrosis and HCC, with elevated serum levels closely related to disease severity (Liu et al., 2022).

Lipid metabolism-related liver diseases

Afamin is a liver-derived glycoprotein indicating disorders associated with abnormal liver lipid metabolism, e.g., increased liver lipid content and insulin resistance. Elevated serum afamin levels in prediabetes and type 2 diabetes patients are closely associated with liver lipid accumulation and liver injury, suggesting afamin’s potential as a biomarker (Kurdiova et al., 2021). Sclerostin, a glycoprotein inhibiting bone formation, is downregulated in NAFLD patients and negatively correlated with metabolic parameters. This downregulation of sclerostin is consistent with increased liver lipid levels, indicating its role in the liver lipid-bone metabolism axis and its practical value as a marker of NAFLD (Zhou et al., 2021).

Although various glycoproteins have been identified as potential biomarkers, their specificities and sensitivities need to be further validated in large-scale, multi-center studies to confirm their clinical values. Additionally, understanding the molecular mechanisms driving the expression of these biomarkers and their roles in the pathogenesis of liver diseases may provide deeper insights into the treatment of lipid metabolism-related liver diseases.

Glycoproteins as targets for drug therapy

The unique structural and functional properties of glycoproteins make them attractive targets for drug therapy, with extensive research in metabolic diseases.

Myocardial infarction and ischemic stroke

Platelet glycoprotein VI (GPVI) is crucial for collagen-induced activation in platelet adhesion and thrombosis in atherosclerosis (Fuentes, 2022). Inhibiting the GPVI signaling pathway represents a promising strategy for antiplatelet therapy, making it a new target aimed at reducing myocardial infarction and ischemic stroke without significantly affecting hemostasis (Borst & Gawaz, 2021). This approach effectively reduces the risk of bleeding complications. Targeted blockage of GPVI has shown efficacy in delaying progressive ischemic brain injury in acute ischemic stroke models and extending the therapeutic window for reperfusion (Bieber et al., 2021). Moreover, overexpression of GPVI dimers in stroke patients further confirms its potential as a therapeutic target (Induruwa et al., 2022).

Central nervous system disorders and HIV-1 infection

Oligodendrocyte myelin glycoprotein (OMGP) has been recognized as a novel antigen in central nervous system disorders such as multiple sclerosis and acute disseminated encephalomyelitis. The presence of anti-OMGP autoantibodies emphasizes its importance in patient stratification and personalized treatment (Gerhards et al., 2020). P-selectin glycoprotein ligand-1 (PSGL-1), as a cell adhesion molecule involved in immune response regulation, has become a potential therapeutic target in HIV-1 infection. New therapeutic strategies targeting PSGL-1 may supplement existing antiretroviral therapies and provide additional benefits in HIV-1 infection control (Zaongo et al., 2021).

Liver metabolic diseases (NAFLD and NASH)

Histidine-rich glycoprotein (HRG), a multifunctional plasma protein produced by hepatocytes, is crucially involved in immune regulation and angiogenesis. HRG’s role has been reported in various experimental models of NAFLD, particularly NASH. During liver ischemia/reperfusion injury (IRI), HRG expression increases and alleviates liver injury by reducing neutrophil infiltration and neutrophil extracellular trap (NET) formation. However, HRG expression is significantly reduced in the steatotic liver, leading to increased susceptibility to IRI (Guo et al., 2021). Parallel findings indicate that HRG supports specific populations of tumor-associated macrophages, playing a key role in NAFLD progression to HCC, indicating HRG as a promising prognostic marker and therapeutic target (Foglia et al., 2024).

As a hepatokine, ZAG exerts protective effects on hepatic steatosis and inflammation. Its expression is decreased in NAFLD, and ZAG overexpression has been shown to alleviate lipid accumulation, improve insulin sensitivity, and inhibit inflammatory responses through various molecular pathways (e.g., IRS/AKT signaling) (Xiao et al., 2018; Liu et al., 2019), positioning ZAG as a promising therapeutic target in NAFLD.

In the context of abnormal hepatic lipid metabolism, changes in drug pharmaco-kinetics also requires attention. In NAFLD, P-glycoprotein (P-gp) expression in the liver is elevated, enhancing bile excretion of drugs such as digoxin, thereby altering their pharmacokinetic characteristics and reducing their bioavailability. This phenomenon must be considered when designing drug therapies for NAFLD cases (Jeong, Lee & Kang, 2021).

α-2-HS glycoprotein (AHSG), also known as fetuin-A, is closely linked to metabolic diseases, e.g., insulin resistance and diabetes mellitus. Melatonin improves insulin resistance and hepatic steatosis by downregulating AHSG and reducing endoplasmic reticulum stress, highlighting the therapeutic potential of targeting fetuin-A in NAFLD (Heo et al., 2018).

Alcoholic liver disease

In diabetic models, reduced β-2 glycoprotein I (β2GPI) protects against liver injury by activating AMPK, subsequently reducing oxidative stress, glucose levels, and inflammatory cytokines (Zhang et al., 2018). Dual-specificity protein phosphatase 1 (Areg), a member of the epidermal growth factor family, protects from immune-mediated acute liver injury by inducing IL-22 production, thereby promoting anti-apoptotic pathways and reducing liver damage (Wu et al., 2020). In alcohol-related liver injury, cellular repressor of E1A-stimulated genes 1 (CREG 1) exerts a protective effect through inhibition of the ASK1-JNK/p38 stress kinase pathway. CREG 1 upregulation also alleviates apoptosis and inflammation, suggesting prospects as a therapeutic target for alcoholic liver disease (Wu et al., 2022).

Existing evidence showcases the therapeutic potential of glycoproteins in lipid metabolism-related liver diseases, emphasizing the importance of continued exploration and development of glycoprotein-targeted therapies to improve the prognosis of affected patients. Despite significant progress, the precise molecular mechanisms by which these glycoproteins affect metabolic liver diseases have yet to be fully elucidated. Understanding these mechanisms is crucial for developing targeted therapies and may help identify new therapeutic targets, providing fresh insights into potential prognostic markers and treatment strategies.

Conclusion

Diseases related to hepatic lipid metabolism encompass metabolic dysfunction-associated fatty liver disease (MAFLD), hyperlipidemia, and hepatic fibrosis, which are associated with abnormal accumulation of lipids and metabolic disorders affecting the liver. Due to their complex pathophysiological features, these diseases are clinically challenging and require further effective treatment options to improve the patient’s quality of life. Accumulating evidence emphasizes glycoproteins’ roles in hepatic lipid metabolism (Fig. 3), underscoring the need for targeted therapeutic strategies. Glycosylation modifications are often altered in diseases such as cancer, autoimmune diseases, or infectious diseases. Glycoproteins exhibit higher sensitivity in certain situations. Changes in glycosylation patterns can amplify disease signals and even be detected in the early stages. However, the sensitivity of traditional biomarkers in early disease stages is usually low, depending on the different glycosylation patterns of specific diseases. Due to the fact that glycoproteins represent more complex molecular changes, glycoprotein markers may also provide higher specificity in certain situations. In addition, glycosylation modifications are dynamic and may vary depending on patient heterogeneity. Therefore, glycoprotein markers have great potential in personalized medicine. Advanced detection methods such as mass spectrometry and glycoproteomics can provide detailed analysis of these markers. Existing evidence showcases the therapeutic potential of glycoproteins in lipid metabolism-related liver diseases, emphasizing the importance of continued exploration and development of glycoprotein-targeted therapies to improve the prognosis of affected patients. Despite significant progress, the precise molecular mechanisms by which these glycoproteins affect metabolic liver diseases have yet to be fully elucidated. Understanding these mechanisms is crucial for developing targeted therapies and may help identify new therapeutic targets, providing fresh insights into potential prognostic markers and treatment strategies.

Figure 3 Glycoproteins are involved in the regulation of hepatic lipid metabolism and related diseases.

SHBG, sex hormone-binding globulin; PEDF, pigment epithelium-derived factor; ZAG, Zinc-α2-glycoprotein; SR-BI, scavenger receptor class B type I; OLFM2, Olfactomedin 2; GPNMB, glycoprotein non-metastatic melanoma B; COMP, cartilage oligomeric matrix protein; HRG, histidine-rich glycoprotein; AHSG, α-2-HS glycoprotein. (Created with MedPeer).

In conclusion, examining the roles of glycoproteins in hepatic lipid metabolism may increase mechanistic understanding regarding the related diseases, besides highlighting potential disease biomarkers and therapeutic targets. On the basis of understanding the mechanism of glycoproteins’ effects on lipid metabolism, further performing advanced glycoproteomic assays to assess liver tissue samples of known pathological features and clinical history may provide a novel avenue for studying diagnostic markers for related diseases, besides increasing our understanding of the complex interactions affecting the glycocalyx during abnormal hepatic lipid metabolism. Future research will continue to explore the mechanism of action of glycoproteins and use this as a basis for developing additional diagnostic and therapeutic approaches to disease.

Additional Information and Declarations

Competing Interests

The authors declare that they have no competing interests.

Author Contributions

Bo Lan analyzed the data, prepared figures and/or tables, and approved the final draft.

Sixing Huang analyzed the data, authored or reviewed drafts of the article, and approved the final draft.

Jinping Luo performed the experiments, prepared figures and/or tables, and approved the final draft.

Ping Shi performed the experiments, prepared figures and/or tables, authored or reviewed drafts of the article, and approved the final draft.

Zhe Qiang conceived and designed the experiments, authored or reviewed drafts of the article, and approved the final draft.

Data Availability

The following information was supplied regarding data availability:

This is a literature review.

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
