# Peer review of "Role of glycoproteins in hepatic lipid metabolism: emerging diagnostic markers and therapeutic targets"

_PeerJ, doi:10.7717/peerj.19627_

## Round 0.1 · original submission · Major Revisions

Please address concerns of all reviewers and amend manuscript accordingly

Reviewer 1 ·

Basic reporting

this study has mentioned many claims that they may not have concern to the objectives of the study or explained such aspect of the study. The report need to be more concise in term of define the term and describing or exploring of the keywords.

there is a title " The audience this review is intended for", which do not make sense. Passage under this title is written to editor not for reader. therefore, all line from 73 to 81, should be subtracted from this manuscript.

Experimental design

Method requires to more detail and correction. The author did not define the design of the study. Most probably this research could be considered as narrative research study. They have to address the reason why systematic review and meta-analysis have not been carry out in this research.
There is some extra claim regarding the important of the study under sub tittle “The audience this review is intended for”, it do not make concern for the study.

Validity of the findings

this study could provide detail information about glycoproteins, while it require to figure and images. The concept has not been cleared, all pathway and mechanism remain ambiguity regarding glycoprotein.
for instance, the author has mentioned that Sex Hormone-Binding Globulin (SHBG) could be used as indicator for many diseases such as alcoholic fatty liver disease (NAFLD) and polycystic ovary syndrome (PCOS). While, clinically there should be a justification to put that in to practice.

·

Basic reporting

This manuscript is well-written and provides a comprehensive overview of glycoproteins in hepatic lipid metabolism. By incorporating the suggested improvements, the paper will be more engaging, visually appealing, and scientifically rigorous, increasing its chances of acceptance in a high-impact journal.

1. Consider making the title more specific and engaging. Like "Glycoproteins in Hepatic Lipid Metabolism: Emerging Diagnostic Markers and Therapeutic Targets or other
2. In the abstract, highlight what makes this review unique compared to existing literature. Conclude with a sentence about the potential impact of glycoprotein research on clinical practice.
3. in Introduction, Provide a brief overview of glycoprotein biology and their general roles in cellular processes before diving into their specific roles in hepatic lipid metabolism. Also, clearly state what is currently unknown or underexplored in the field, which this review aims to address.

Experimental design

Strengths: The methodology section is clear and outlines the search strategy and inclusion criteria.

Suggestions:
1. Include a flow diagram to visually represent the literature search and selection process.
2. Explain why studies from 2012 onwards were selected (e.g., significant advances in glycoprotein research during this period).
3. Briefly state why certain articles were excluded (e.g., non-English articles, low relevance).
4. Include diagrams or pathways showing how glycoproteins interact with lipid metabolism pathways in the liver.
5. Emphasise how these findings could translate into clinical applications (e.g., diagnostic tests, targeted therapies).
6. Discuss how glycoprotein-based markers compare to traditional biomarkers in terms of sensitivity, specificity, and clinical utility.
7. In section Glycoproteins as Targets for Drug Therapy, group therapeutic targets by the diseases they target (e.g., NAFLD, HCC, cardiovascular diseases). In addition, address potential challenges in targeting glycoproteins (e.g., off-target effects, delivery methods).

Validity of the findings

No any

·

Basic reporting

No comments

Experimental design

No comments

Validity of the findings

No comments

Additional comments

This review article open a new direction for further research in this area

---

## Round 0.2 · accepted · Accept

All issues pointed by the reviewers were addressed and the manuscript is acceptable now.